# Development of Quantitative Real-Time PCR and Loop-Mediated Isothermal Amplification Assays for the Surveillance and Diagnosis of Herpes B Virus Infection

**DOI:** 10.3390/v15102086

**Published:** 2023-10-13

**Authors:** Murasaki Amano, Krittiga Sapkanarak, Wipaporn Thbthimthong, Suthirote Meesawat, Taratorn Kemthong, Nutchanat Suttisan, Haruka Abe, Suchinda Malaivijitnond, Jiro Yasuda

**Affiliations:** 1Department of Emerging Infectious Diseases, National Research Center for the Control and Prevention of Infectious Diseases (CCPID), Nagasaki University, Nagasaki 852-8523, Japan; bb55420102@ms.nagasaki-u.ac.jp (M.A.); abeh@nagasaki-u.ac.jp (H.A.); 2Program for Nurturing Global Leaders in Tropical and Emerging Communicable Diseases, Graduate School of Biomedical Sciences, Nagasaki University, Nagasaki 852-8523, Japan; 3Department of Emerging Infectious Diseases, Institute of Tropical Medicine, Nagasaki University, Nagasaki 852-8523, Japan; 4National Primate Research Center of Thailand, Chulalongkorn University, Saraburi 18110, Thailand; krittiga.s@chula.ac.th (K.S.); wipaporn.t@chula.ac.th (W.T.); suthirote.m@gmail.com (S.M.); taratorn.k@chula.ac.th (T.K.); nutchanatbell@gmail.com (N.S.); suchinda.m@chula.ac.th (S.M.); 5Department of Biology, Faculty of Science, Chulalongkorn University, Bangkok 10330, Thailand; 6Vietnam Research Station, Institute of Tropical Medicine (NEKKEN), Nagasaki University, Nagasaki 852-8523, Japan

**Keywords:** herpes B virus, zoonosis, LAMP, surveillance, cynomolgus macaque

## Abstract

Herpes B virus (BV) is a zoonotic virus which can be transmitted from macaques to humans, which is often associated with high mortality rates. Because macaques often exhibit asymptomatic infections, individuals who come into contact with these animals face unexpected risks of BV infections. A serological test is widely performed to investigate BV infections. However, the assay’s sensitivity and specificity appeared to be inadequate, and it does not necessarily indicate ongoing viral shedding. Here, we developed LAMP and qPCR assays aiming to detect BVs with a high sensitivity and specificity in various macaque species and validated them using oral swab samples collected from 97 wild cynomolgus macaques living in Thailand. Our LAMP and qPCR assays detected more than 50 and 10 copies of the target sequences per reaction, respectively. The LAMP assay could detect BV within 25 min, indicating its advantages for the rapid detection of BV. Collectively, our findings indicated that both assays developed in this study exhibit advantages and usefulness for BV surveillance and the diagnosis of BV infections in macaques. Furthermore, for the first time, we determined the partial genome sequences of BVs detected in cynomolgus macaques in Thailand. Phylogenetic analysis revealed the species-specific evolution of BV within macaques.

## 1. Introduction

Herpes B virus (*Macacine alphaherpesvirus* 1; BV) belongs to the genus *Simplexvirus*, subfamily *Alphaherpesvirinae*, and is a zoonotic pathogen that causes a life-threatening disease in humans. The natural hosts of BV are macaques, and BV infection is usually asymptomatic in natural hosts. BV can be transmitted from infected macaques to humans via direct contact with body fluids, scratches, or bites, and results in fatal human central nervous system diseases, including encephalitis and encephalomyelitis [1]. Macaque monkeys are abundant in Southeast Asian countries. In Thailand, the cynomolgus macaque or long-tailed macaque (*Macaca fascicularis*) and rhesus macaque (*Macaca mulatta*) are widely distributed and reside close to human residential areas [2], thereby posing a significant risk to public health. BV persists in trigeminal nerves and spinal ganglia of macaques throughout their lives. Viral shedding can potentially occur, especially in conjunction with immunosuppression, breeding season, stress, and primary infection [3]. The virus can be contained in saliva, urine, and genitalia during the viral reactivation [4]. A previous study has shown that seropositivity was more than 50% in rhesus macaques and 60% in cynomolgus macaques reared in laboratory animal facilities in Japan [5]. Moreover, 31 out of 38 (81.6%) wild adult rhesus macaques were BV seropositive in Bali [6]. Regardless of their habitats, the BV seroprevalence of adult macaques was higher than that of the young macaques [6]. Recently, the use of non-human primates for experiments has been gradually increasing every year for the evaluation of the efficacies of candidate drugs or vaccines. This is primarily due to the genetic, physiological, and immunological characteristics shared between humans and primates [7]. Therefore, workers who handle macaque monkeys are at a high risk of acquiring these viruses. Indeed, on 30 June 2021, a human infection case involving BV was reported in China. The patient eventually died two months after dissecting dead monkeys in a research institute [8], highlighting the importance of reliable BV detection before performing procedures with monkeys. Misdiagnosis and delay in treatment with patients may lead to fatal diseases in humans.

Thus, to evaluate the risk of BV transmission from macaques to humans in areas where humans and macaques live close, such as laboratory animal facilities, assays that can detect BV with a high accuracy and sensitivity can be used for the required surveillance and regular inspection of BV. At present, the standard assay used for the detection of BV involves viral isolation from clinical samples. However, a biosafety level (BSL)-3 or BSL-4 containment facility is required to propagate BV in cell lines, such as Vero cells and Hela cells [9,10]. Moreover, it takes approximately one to seven days to isolate BV [11]. Although an enzyme-linked immunosorbent assay (ELISA) has been commonly used in primate facilities as the serological method to detect antibodies against BV, it can detect only a history of BV infection but does not provide information on the shedding of the virus. Furthermore, detecting antibodies against BV in cases of recent infections is challenging, and there is a cross-reactivity between BV and other herpesviruses, including Herpes Simplex Virus (HSV) 1 and 2 [12]. A quantitative real-time PCR (qPCR) may be the most sensitive and accurate assay to diagnose BV infection. However, it has not gained widespread adoption as a surveillance assay, primarily due to the requirement for sophisticated instrumentation and stable electric power. In addition, previously reported primers for qPCR were designed based on the BV E2490 strain isolated from rhesus macaque [13], suggesting that it may be challenging to identify BV from other macaque species [14]. Therefore, a simple and rapid BV detection method is essential. One of the alternative methods to qPCR is the loop-mediated isothermal amplification (LAMP) assay, which amplifies the target DNA without thermal cycling steps. LAMP uses four to six primers, which increases the efficiency of target genome amplification [15]. We developed rapid and sensitive LAMP assays for the detection of SARS-CoV-2, ebolaviruses, Marburgvirus, Zika virus, Lassa virus, foot and mouth disease, and bovine papular stomatitis virus [16,17,18,19,20,21,22,23,24,25,26]. Additionally, the LAMP assay is a simple technique with battery-powered equipment which does not require other advanced equipment and can be made available in mobile clinics, as well as remote areas [18,19,27].

In this study, we developed LAMP and qPCR assays which can be employed to detect all BV strains identified so far. Both assays were validated using samples collected from wild and captive cynomolgus macaques in Thailand.

## 2. Materials and Methods

### 2.1. Ethics Statement

Sample collection from wild cynomolgus macaques was approved by the Department of National Parks, Wildlife, and Plant Conservation of Thailand and the Institutional Animal Care and Use Committee (IACUC) of the National Primate Research Center of Thailand-Chulalongkorn University (Protocol Review Nos. 2075007 and 2375008).

### 2.2. Sample Collection and DNA Extraction

Ninety-seven cynomolgus monkeys were captured from the central and southern areas of Thailand, and oral swab and blood samples were collected, and the animal health was checked by complete blood test using blood samples. Regarding oral swab samples, the oral cells and saliva were harvested from the inside of both cheeks, the base of the tongue and the soft palate. To minimize animal distress, anesthetics with 2–5 mg/kg body weight of Tiletamine HCl and zolazepam HCl (Virvac, Nice, France) were intramuscularly administered to all monkeys during specimen collection. All swabs were kept in a screw-cap tube containing a DNA/RNA shield (ZYMO Research, Irvine, CA, USA) to stabilize the DNA until use. Following the manufacturer’s protocol, whole DNA was extracted from swab samples using a QIAamp DNA Mini Kit (QIAGEN, Hilden, Germany). The final elution volume was 150 µL.

### 2.3. Preparation of the BV DNA Fragments

Synthesized DNA based on the sequences of 248 bp of UL27 (1850–2097), 164 bp of gG (303–466), and 257 bp of UL29 (1093–1349) were purchased from Eurofins Genomics, Tokyo, Japan. Plasmids containing these BV sequences were constructed using a Zero Blunt TOPO PCR Cloning Kit (Invitrogen, Waltham, MA, USA). Plasmids were digested with *Eco*RI. DNA fragments were electrophoresed on agarose gels and purified using a QIAquick Gel Extraction Kit (QIAGEN). The purified DNAs were serially diluted with distilled water from 1 × 10^6^ to 1 × 10^0^ DNA copies/µL.

### 2.4. qPCR for Validation of Viral DNA Quantity

The qPCR primer sequences are listed in Table 1. Primer sequences were based on the conserved sequences of the BV strains (accession No. KY628968-77, KY628979-85, and KJ566591). The TaqMan probe for BV detection was fluorescently labeled with 6-carboxyfluorescein (FAM) at the 5′ ends and 6-carboxytetramethylrhodamine (TAMRA) at the 3′ ends. All probes were synthesized by Eurofins Genomics (Japan). qPCR was performed using a QuantiStudio™ 5 qPCR system (Applied Biosystems, Waltham, MA, USA). The reaction mixture (10 µL) contained 5 µL of 2× TaqMan™ Fast Advanced Master Mix; 1 µL of 10× primer–probe mixture, including 9 µM forward and reverse primers, and 2 µM probe, 3 µL of water, and 1 µL of extracted DNA. The reaction was conducted with a thermal cycle program of 50 °C for 2 min, 95 °C for 20 s, followed by 40 cycles of 95 °C for 1 s, and 60 °C for 20 s. The cut-off values were set at the threshold cycle (Ct) value of 40. To quantify the viral DNA, a standard curve was created with 10-fold serial dilutions of the purified BV DNA fragment possessing the qPCR target sequence. The assay was performed in duplicate for each sample.

### 2.5. LAMP Primer Design

LAMP primers were designed based on the sequences of the conserved regions among the BV strains. The whole BV genome sequence was aligned using the MAFFT version 7 online software (https://mafft.cbrc.jp/alignment/server/, accessed on 30 July 2023) to identify conserved regions. Primer Explorer V5 software (Eiken, Shizuoka, Japan; http://primerexplorer.jp/, accessed on 30 July 2023) was used to design LAMP primers. The LAMP method requires a set of six primers (F3, B3, FIP, BIP, LF, and LB) that recognize eight different sites in the target DNA.

### 2.6. LAMP Reaction

LAMP assays were performed using a Genelyzer FIII real-time fluorescence detection platform (Canon Medical Systems, Otawara, Japan), as previously described [16]. LAMP reaction mixture (25 µL) included 15 µL of Isothermal Master Mix (ISO004, Canon Medical Systems), 0.8 µM each of FIP and BIP primer, 0.4 µM each of F and B loop primers, 0.2 µM each of F3 and B3, 2.5 µL of water, and 5 µL of each sample. The reaction was performed at 65 °C for 40 min, followed by a dissociation analysis at 95–75 °C with a temperature change rate of 0.1 °C/s. Non-specific positive samples were excluded by comparing the melting temperature to that of the positive control. The sample was considered positive when its fluorescence intensity was >20,000 within 30 min of amplification. The assay was performed in duplicate for each sample.

### 2.7. Sanger Sequencing to Confirm the Conserved Primer Regions among BV Strains

To examine whether the viral genome sequences of primer target regions are conserved, partial genomes, including the primer binding sites of BV-positive samples, were amplified by PCR using the primer sets shown in Appendix A. Nested-PCR was performed using PrimeSTAR GXL DNA Polymerase (Takara Bio, Kusatsu, Japan) under the following conditions: 30 cycles each of 10 s at 98 °C, 15 s at 55 °C, and 1 min/kb at 68 °C. All the PCR products were electrophoresed on agarose gels and purified from the gels using the QIAquick Gel Extraction Kit. The purified DNAs were processed using the BigDye Terminator v3.1 Cycle Sequencing kit (Thermo Fisher Scientific, Waltham, MA, USA) and analyzed with the SeqStudio Genetic Analyzer. The collected sequences were identified using BLAST searches (https://blast.ncbi.nlm.nih.gov, accessed on 30 July 2023). The obtained sequences were deposited in the DNA Data Bank of Japan (DDBJ) database under the following accession numbers: LC777613–LC777618.

### 2.8. BV Phylogenetic Analysis

To characterize the BV strains, No. 8-14 and No. 8-17, identified in this study, the region of gJ (US5) to gD (US6) genes (720 or 723 bp), which is a hypervariable region of the BV genome, was amplified by PCR using the primer sets shown in Appendix A. The PCR products were purified from the gels and then sequenced as described above. The phylogenetic analysis of the sequence data was performed using BioEdit version 7.2.5 and MEGA 7 software. A phylogenetic tree was constructed using the neighbor-joining method. The neighbor-joining tree was constructed at 1000 bootstraps. The sequence data of No. 8-14 and No. 8-17 were deposited in DDBJ under the accession numbers LC777619 and LC777620, respectively.

## 3. Results

### 3.1. Development of a Novel qPCR Assay to Detect BV

The qPCR method for detecting BV has been previously reported by Perelygina et al. [14]. The primer set for qPCR (hereafter called qPCR-gG) was designed based on the glycoprotein G (gG) region sequence of the BV strain E2490 isolated from the rhesus macaque (*Macaca mulatta*) (Table 1). We analyzed the sequences of the binding region of the qPCR-gG primers in all the BV strains deposited in GenBank, including the strains isolated from other macaque species; KY628982 from Japanese macaque (*Macaca fuscata*), KY628968 from lion-tailed macaque (*Macaca silenus*), KY628985 bonnet macaque (Macaca radiata), and KY628969-70 from southern pig-tailed macaque (Macaca nemestrina), and found a nucleotide mismatch at the 3′ end of the forward primer (G to C) and the probe (G to C) in the BV strains isolated from cynomolgus macaques (*Macaca fascicularis*) (Figure 1). The mismatches at the 3′ end of primers plausibly affected the sensitivity of the assay. Therefore, to detect all the known BV strains with high sensitivity, we designed a new qPCR primer set targeting the UL29 region, which is conserved among all the BV strains (Figure 2). The mixed nucleotide probe was used in the qPCR because a single nucleotide diversity (A or G) existed at the sixth nucleotide of the probe-binding region between the strains isolated from the cynomolgus and rhesus macaques.

We compared the sensitivities of qPCR-gG and the newly designed qPCR (qPCR-UL29) using plasmids containing the primer-binding sequences of the E2490 strain isolated from rhsus macaques or the E90-136 strain isolated from cynomolgus macaques. The detection limit of qPCR-gG was 10 copies or 100–1000 copies for E2490 and for E90-136, respectively, whereas that of qPCR-UL29 was 10 copies for both E2490 and E90-136 (Table 2). Thus, qPCR-UL29 was more sensitive than qPCR-gG for the detection of E90-136, although both qPCR assays showed similar sensitivities for the detection of E2490. This suggests that qPCR-UL29 is more suitable for detecting a broad range of BV strains.

### 3.2. Evaluation of the qPCR Assay Using Monkey Specimens

To further evaluate the qPCR assay for BV detection, 97 oral swab samples from the cynomolgus macaques were examined using qPCR-gG. The result showed that 19 samples were positive (Table 3). Subsequently, we evaluated the qPCR-UL29 assay using the same 97 oral specimens. The assay detected BV in only two samples, No. 8-14 and No. 8-17, which had shown positive results in qPCR-gG (Table 3). Positive samples in qPCR-gG were further confirmed by employing a nested PCR assay using an in-house primer set (Appendix A) since qPCR-gG and qPCR-UL29 showed different results. The amplicon length expected after specific amplification by nested PCR is 359 bp. The amplification products with expected size were detected in only two samples, No. 8-14 and No. 8-17, among nineteen qPCR-gG-positive samples (Appendix A). Taken together with the result from qPCR-UL29, we concluded that the remaining 17 samples other than No. 8-14 and No. 8-17 were false positives. Although the amplification curves of false positive samples in qPCR-gG were similar to those of No. 8-14 and No. 8-17 (Appendix A), they were not specific amplifications. The Ct values of No. 8-14 and No. 8-17 were 38.3 (1.25 copies) and 33.0 (15.95 copies) in qPCR-gG and 28.0 (12,748 copies) and 24.6 (177,970 copies) in qPCR-UL29, respectively (Table 3). This confirmed the fact that qPCR-UL29 exhibits a high sensitivity and specificity in comparison to qPCR-gG.

### 3.3. Development of the LAMP Assay for BV

To establish a rapid and sensitive detection assay for BV, we developed the LAMP assay. LAMP primers were designed based on sequences within the UL27 region, which is conserved among BV strains. Among the primers, only the B3 primer had one nucleotide mismatch with the sequences of BV isolates other than the rhesus macaque (Figure 3). However, this mismatch exhibited a minimal effect on the sensitivity because of its location near the 5′ terminus (Figure 3).

The LAMP assay was performed using DNA fragments containing primer-binding sequences of the E2490 strain isolated from rhesus macaques or the E90-136 strain isolated from cynomolgus macaques. As shown in Table 4, the detection limit of the LAMP assay was 50 copies per reaction for both E2490 and E90-136. The assay detected more than 50 copies of the E2490 and E90-136 plasmid within 25 and 19 min, respectively, indicating that the LAMP assay exhibited a rapid and high sensitivity for the detection of BV.

### 3.4. Validation of the LAMP Assay Using Monkey Specimens

To validate the LAMP assay, 97 oral swabs of wild cynomolgus macaques determined in the qPCR assays mentioned above were used (Table 5). Only two samples, No. 8-14 and No. 8-17, were detected to be positive in the LAMP and qPCR-UL29 assays, indicating that the LAMP assay results were consistent with those of the qPCR-UL29 assay (Table 6). The average time for the detection of positivity was 13 min 8 s and 9 min 54 s for No. 8-14 and No. 8-17, respectively.

### 3.5. Sequence Analyses for the Primer Binding Sites

To investigate whether the primer sequences had mismatched sequences with the selected genes of the BV strains from the specimens of Thai cynomolgus macaques, we determined the sequences of the primer binding sites of No. 8-14 and No. 8-17. The qPCR-UL29 primers did not exhibit a mismatch in both No. 8-14 and No. 8-17 strains (Figure 2), while the LAMP primers possessed two nucleotide mismatches at the sixth position from the 5′ terminus of the F3 primer and fifth position from the 3′ terminus of the LF primer in both No. 8-14 and No. 8-17 strains (Figure 3). In addition, the No. 8-14 strain exhibited another mismatch at the eighth position from the 3′ terminus of the B2 primer region. Therefore, the DNA fragment containing the sequence of the primer-binding region of No. 8-14 was prepared as described in the Materials and Methods section and used to evaluate the sensitivity of the LAMP assay. The detection limit was determined to be 50 copies (Table 7).

### 3.6. Phylogenetic Analysis of BV Strains Isolated from Thai Cynomolgus Macaques

In this study, we detected BV in two Thai cynomolgus macaques. To date, genomic data on BV in macaques living in Southeast Asian countries have not been reported. Additionally, BV genomic data from cynomolgus macaques are limited, even globally. To characterize the BV strains detected in this study, we determined the sequences of gJ (US5) through gD (US6) genes (720 or 723 bp), which is a hypervariable region of the BV genome [28], and performed phylogenetic analysis. The phylogenetic tree showed three distinct lineages: rhesus and Japanese macaques (*M. mulatta* and *M. fuscata*), cynomolgus macaques (*M. fascicularis*), and pig-tailed macaques (*M. nemestrina*). BV strains derived from the Thai cynomolgus macaques from our study clustered with the cynomolgus BV strain E90-136, as previously reported (accession No. AF082811; Figure 4). Among our cynomolgus macaque’s BV strains, the No. 8-14 strain was slightly different from the No. 8-17 strain, which was clustered with the BV strain E90-136. This might be because these BV strains were isolated from different subspecies of cynomolgus macaques; the No. 8-14 strain was from the common cynomolgus macaque (*M. fascicularis fascicularis*) and the No. 8-17 strain was from the Burmese cynomolgus macaque (*M. fascicularis aurea*). Unfortunately, we do not know the subspecies level of cynomolgus macaques infected with the E90-136 strain. It is noted that the BV genomes from cynomolgus macaques were entirely different from the larger clade of BV strains of rhesus and Japanese macaques.

## 4. Discussion

In this study, we developed a qPCR-UL29 assay that can efficiently detect BVs originating from both rhesus and cynomolgus monkeys, as well as a LAMP assay that can rapidly and specifically detect BVs. Both assays were practical for BV detection in macaque oral specimens. Although the multiplex LAMP assay, which can detect both BV and Mpox virus simultaneously, was recently reported to demonstrate a high specificity and sensitivity for the detection of BV, it was evaluated only using samples containing BV plasmid in BV-negative monkey serum due to the limited availability of monkey specimens [29]. To the best of our knowledge, this is the first study to evaluate a LAMP assay for BV using monkey oral samples. The results of the LAMP assay were consistent with those of the qPCR assay, suggesting that the LAMP assay was a reliable method for BV detection.

The detection limit of the LAMP assay was 50 copies/reaction, whereas that of the qPCR-UL29 assay was 10 copies/reaction (Table 2 and Table 4), indicating that the qPCR-UL29 assay exhibited a five-fold higher sensitivity than the LAMP assay. This may be due to the high GC content of the BV genome (74% G + C composition), which plausibly impacts the efficiency of the LAMP amplification. In the complicated reaction steps of the LAMP, GC-rich targets may cause unexpected dimers between the primers themselves or between the primers and amplicons. According to a previous study, viral shedding of HSV-2 frequently occurred at rates of 10^5.2^ copies/mL (10^2.2^ copies/µL) and exhibited high infectivity [30,31]. Therefore, our LAMP assay displays adequate sensitivity to detect a transmissible number of BV. The LAMP assay has several advantages, including simple experimental procedures, a faster detection rate than the qPCR, and the use of portable equipment [32], suggesting that the LAMP assay proves to be a valuable method for point-of-care diagnosis and field surveillance.

In this study, we identified two BV-positive samples, No. 8-14 and No. 8-17, in oral swab specimens from wild cynomolgus monkeys living in Thailand. The detection limit of the assay was 50 copies, which is comparable to a recent report [29], implying high sensitivity in the case of monkey specimens. Although the BV genome from cynomolgus macaques in Thailand had a sequence mismatch in the F3, LF, and BIP primer regions (Figure 3), the BV genome was appropriately detected in sample No. 8-14 as positive by our LAMP assay. A recent report showed that a sequence mismatch located near the 3′ end of the FIP and BIP primers affects the sensitivity of the assay [33]. However, the sequence mismatch of primers had minimal effect on the sensitivity of our LAMP assay, since our primer set did not exhibit a mismatch near the 3′ end of FIP and BIP.

Regarding the qPCR assay, the sequences of the qPCR-UL29 primers had no mismatch with the BV 8-14 and 8-17 strains identified in Thai cynomolgus macaques in this study and all the BV strains collected from at least seven macaque species of which the genome sequences have been registered in GenBank so far (Figure 2), suggesting that the qPCR-UL29 assay could be applied to most species of macaques.

BV is an important zoonotic disease that is transmitted from macaques to humans and shows a high mortality rate. In Southeast Asian countries, macaques live close to human residential areas in many towns. The transmission of BV from wild macaques in these towns and areas is a serious public health concern. Therefore, BV surveillance in macaques would be necessary in the context of “One Health”.

Collectively, our findings indicated that the qPCR and LAMP assays developed in this study are useful for the diagnosis of BV for any macaque monkeys and the surveillance of BV in monkey populations. In addition, our LAMP assay would be applicable to human specimens as a diagnostic assay for BV infection, since the target sequences of the LAMP primers are less conserved among human herpes simplex virus (HSV)-1 and HSV-2 strains (Appendix A).

Furthermore, we determined the partial genome sequences of two BV strains, No. 8-14 and No. 8-17, identified in Thai cynomolgus macaques. This is the first report of BV genome sequences from Thai cynomolgus macaques. Phylogenetic analysis showed that both the Thai strains belonged to the same clade as the E90-136 strain, which had previously been isolated from cynomolgus macaques reared in a primate facility in the US. Interestingly, the two BV strains (Nos. 8-14 and 8-17) were separated into different clades based on the host subspecies level (*M. fascicularis fascicularis* and *M. fascicularis aurea*). Recently, it was reported that the genetic characteristics, based on genetic markers of mtDNA, Y chromosome genes *SRY* and *TSPY*, whole genome sequences, and autosomal SNPs, are very distinctive between these two subspecies [34,35,36,37]. This reflects that each clade of BVs has independently evolved in a specific macaque species/subspecies [38]. This was also confirmed by the bigger picture that *M. fascicualris*, *M. fuscata,* and *M. mulatta*—which belonged to the same species group of *fascicularis*—were separated from *M. nemestrina,* which belonged to the *silenus-sylvanus group* [39]. Thus, the susceptibility and the co-evolution of the host and BV’s genetics should be taken into account.

## Figures and Tables

**Figure 1 viruses-15-02086-f001:**
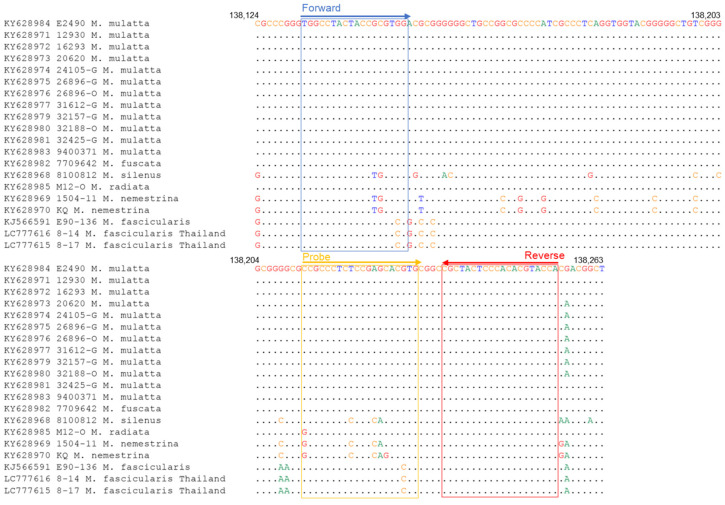
Sequence alignments of qPCR-gG primer target region of BV genome. The accession number of the genome sequence and the macaque origin of each BV strain are shown on the left.

**Figure 2 viruses-15-02086-f002:**
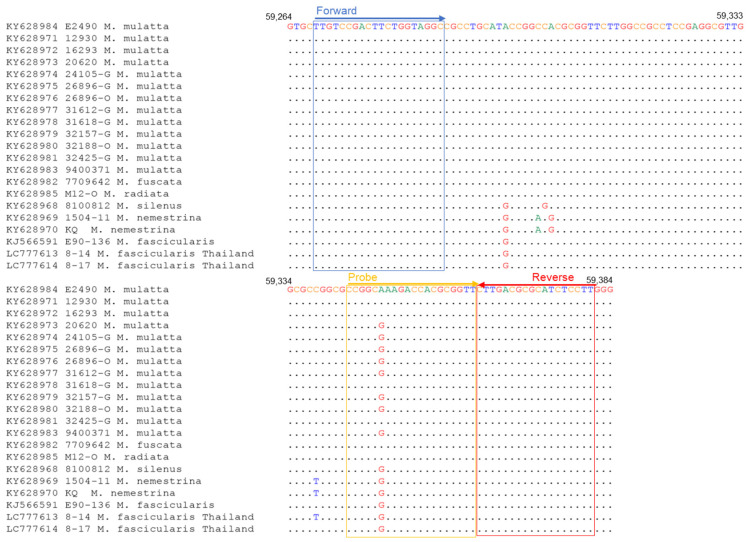
Sequence alignments of qPCR-UL29 primer target region of BV genome. The accession number of the genome sequence and the macaque origin of each BV strain are shown on the left.

**Figure 3 viruses-15-02086-f003:**
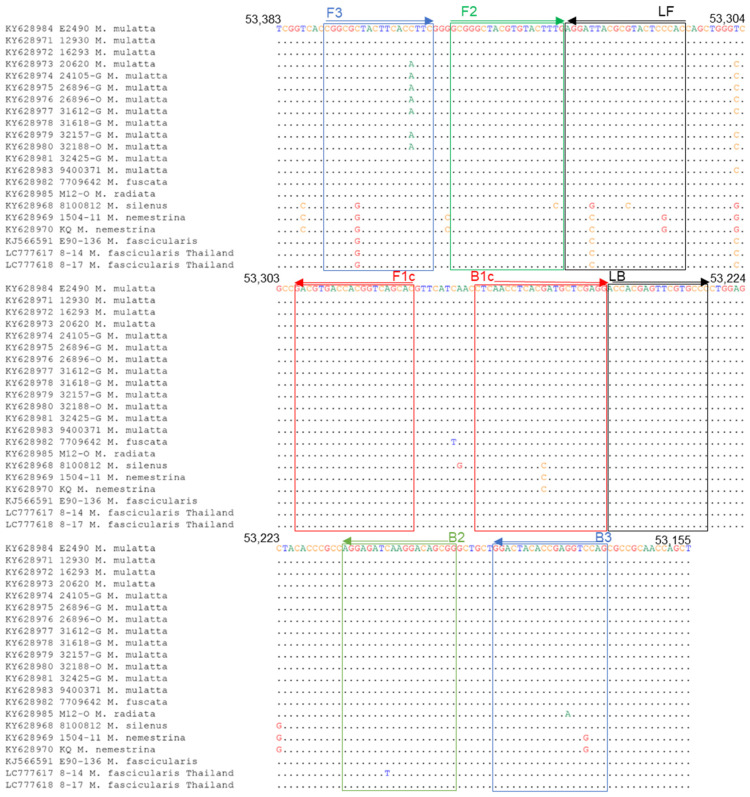
Sequence alignments of LAMP primer target regions of BV genome. The accession number of the genome sequence and the macaque origin of each BV strain are shown on the left.

**Figure 4 viruses-15-02086-f004:**
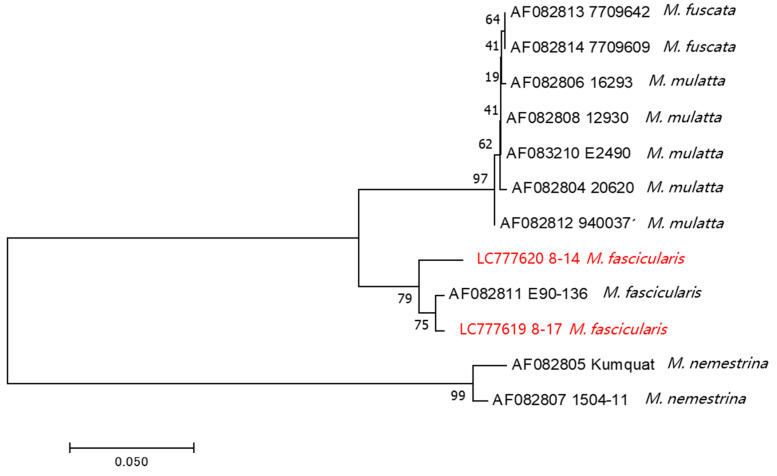
Phylogenetic analysis for the BV strains from cynomolgus macaques in Thailand and other BV strains. LC777619 and LC777620 are the BVs identified in this study (red color). The US5 (gJ) open reading frame (ORF) to US6 (gD) ORF sequence is amplified by PCR, and the product has been sequenced. The neighbor-joining tree is constructed at 1000 bootstraps.

**Table 1 viruses-15-02086-t001:** The sequences of qPCR and LAMP primers.

Primer Set	Primer	Sequence (5′-3′)
qPCR-gG	BV-qPCR-gG-323_Fw	TGGCCTACTACCGCGTGG
	BV-qPCR-gG-446_Rv	TGGTACGTGTGGGAGTAGCG
	BV-qPCR-gG-403_Probe	CCGCCCTCTCCGAGCACGTG
qPCR-UL29	BV-qPCR_UL29_Fw	TTGTCCGACTTCTGGTAGGC
	BV-qPCR_UL29_Rv	AAGGAGATGCGCGTCAAG
	BV-qPCR_UL29_Probe	CCGGCRAAGACCACGCGGTT
BV-gG_1st	BV-gG_Fw_1	GACCCCGCGTACTGCTAC
	BV-gG_Rv_1	CCCACCAGGATCTCGTAGTC
BV-gG_nested	BV-gG_Fw_2	GCCGAYGTCGACAGACAT
	BV-gG_Rv_1	CCCACCAGGATCTCGTAGTC
LAMP-UL27	LAMP_UL27_F3	CGGCGCTACTTCACCTTC
	LAMP_UL27_B3	GGACCTCGGTGTAGTCCAG
	LAMP_UL27_FIP	GTGCTGACCGTGGTCACGTCGCGGGCTACGTGTACTTTG
	LAMP_UL27_BIP	CTCAACCTCACGATGCTCGAGGCCGCTGTCCTTGATCTCCT
	LAMP_UL27_LF	GTGGGAGTACGCGTAATCCT
	LAMP_UL27_LB	ACCACGAGTTCGTGCCC

**Table 2 viruses-15-02086-t002:** Evaluation of qPCR-gG and qPCR-UL29 assays using DNA fragments based on either the E2490 or E90-136 strain.

DNA	E2490						E90-136					
qPCR	qPCR-gG			qPCR-UL29			qPCR-gG			qPCR-UL29		
No. of DNA Copy	Ct Value		Positive	Ct Value		Positive	Ct Value		Positive	Ct Value		Positive
1 × 10^6^	21.16	21.18	2/2	20.68	20.37	2/2	23.16	23.01	2/2	18.99	18.96	2/2
1 × 10^5^	24.65	24.47	2/2	23.33	22.88	2/2	27.49	27.57	2/2	22.38	22.5	2/2
1 × 10^4^	27.91	27.71	2/2	26.33	26.2	2/2	30.48	30.46	2/2	25.77	26.1	2/2
1 × 10^3^	31.21	30.93	2/2	29.76	29.7	2/2	33.28	33.76	2/2	29.36	30.31	2/2
1 × 10^2^	34.89	34.74	2/2	33.12	33.06	2/2	39.3	-	1/2	33.17	33.3	2/2
1 × 10^1^	37.52	37.12	2/2	38.49	38.3	2/2	-	-	0/2	38.58	38.3	2/2
1 × 10^0^	-	-	0/2	-	-	0/2	-	-	0/2	-	-	0/2

**Table 3 viruses-15-02086-t003:** The results of qPCR-UL29 and nested PCR-gG for qPCR-gG positive samples.

Sample ID	qPCR-gG		qPCR-UL29		Nested PCR-gG
	Average of Ct Value	Estimated DNA Copy Number	Average of Ct Value	Estimated DNA Copy Number	
8-1	36.17	3.23	-	-	-
8-2	36.93	1.99	-	-	-
8-3	36.39	2.81	-	-	-
8-4	34.95	7.02	-	-	-
8-5	35.78	4.14	-	-	-
8-6	38.14	0.92	-	-	-
8-7	37	1.91	-	-	-
8-8	36.43	2.74	-	-	-
8-9	36.67	2.35	-	-	-
8-10	37.85	0.58	-	-	-
8-11	37.11	0.95	-	-	-
8-12	37.58	0.69	-	-	-
8-13	37.36	0.8	-	-	-
8-14	38.3	1.25	28	12,748	+
8-15	36.95	1.06	-	-	-
8-16	36.11	1.87	-	-	-
8-17	33	15.95	24.6	177,970	+
8-18	35.14	3.56	-	-	-
8-19	35.58	2.65	-	-	-

**Table 4 viruses-15-02086-t004:** Sensitivity of LAMP assay using DNA fragments based on either the E2490 or E90-136 strain.

DNA	E2490		E90-136	
DNA Copies/ Reaction	Amplification Time (mm:ss)		Amplification Time (mm:ss)	
	Run 1	Run 2	Run 1	Run 2
5 × 10^6^	6:46	6:31	8:16	7:46
5 × 10^5^	7:16	7:16	9:31	9:01
5 × 10^4^	12:01	8:16	10:31	10:16
5 × 10^3^	15:16	9:31	12:31	12:16
5 × 10^2^	15:16	11:16	19:16	16:16
5 × 10^1^	24:16	21:31	17:46	18:16
5 × 10^0^	-	-	-	-

**Table 5 viruses-15-02086-t005:** Evaluation of the qPCR-UL29 and LAMP assays using monkey samples.

		LAMP Assay	
		No. of Positive Samples	No. of Negative Samples
qPCR assay	No. of positive samples	2/97	0/97
	No. of negative samples	0/97	95/97

**Table 6 viruses-15-02086-t006:** The BV detection time of LAMP assay in positive samples.

Sample ID	LAMP-UL27		
	Amplification Time (mm:ss)		Average (mm:ss)
8-14	12:31	13:46	13:08
8-17	9:46	10:01	9:54

**Table 7 viruses-15-02086-t007:** The evaluation of LAMP sensitivity using fragmented DNA based on No. 8-14 strain.

DNA	DNA Based on No. 8-14	
DNA Copies/Reaction	Amplification Time (mm:ss)	
	Run 1	Run 2
5 × 10^6^	8:31	8:46
5 × 10^5^	10:01	10:13
5 × 10^4^	11:46	11:16
5 × 10^3^	14:01	13:46
5 × 10^2^	17:31	16:46
5 × 10^1^	22:01	-
5 × 10^0^	-	-

## Data Availability

All available data are presented in the article.

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
