# Peer review of "Development of Quantitative Real-Time PCR and Loop-Mediated Isothermal Amplification Assays for the Surveillance and Diagnosis of Herpes B Virus Infection"

_viruses, 2023, doi:10.3390/v15102086_

Round 1

Reviewer 1 Report

The reviewed manuscript is dedicated to the design and validation of a qPCR and LAMP assay for herpes B virus, a causative agent of infection in macaques and humans. The presented results are interesting for scientists, specializing on the field of molecular diagnostics. However, a number of issues needs to be addressed before publication.

 Major issues:

1.      Supplementary Figure 1 — there are faint bands between 300-400 bp in samples 8-3 and 8-5. Authors are encouraged to provide qPCR amplification curves for false-positive samples in separate supplementary figures. Observed Cq for qPCR-gG could be artifacts or linear increase of fluorescence.

2.      Authors mentioned that a considerable percent of macaques is BV-seropositive. However, using oral swabs only 2 out of 97 animals were positive after qPCR and LAMP testing. Thus, a question arises about seropositivity of the studied macaques and were there any infection symptoms false-positive and true-positive samples macaques? It can be assumed that much more animals were actually infected but were tested as negative because the infection did not clinically manifest at the time when samples were taken. In that sense, usage of NAATs for macaques can be limited and serological analysis would provide more reliable information.

Minor issues:

1.      Minor language issues need to be corrected.

2.      Page 3, line 127: “0.9 µM forward and reverse primers and 0.2 µM probe” — plausibly, these are not 10x but final concentrations.

3.      Page 3, line 140: “that recognize six different sequences” — FIP and BIP primers are complementary each to 2 different sites.

4.      Page 4, line 191: “The mixed nucleotide probe was used in the qPCR” — it could be seen on Figure 2 that a conservative region without any polymorphisms is located upstream of the designed probe. Plausibly, this region would be more beneficial for probe design.

A few language issues need to be corrected such as typos and douple spaces.

Author Response

Comments and Suggestions for Authors

The reviewed manuscript is dedicated to the design and validation of a qPCR and LAMP assay for herpes B virus, a causative agent of infection in macaques and humans. The presented results are interesting for scientists, specializing on the field of molecular diagnostics. However, a number of issues needs to be addressed before publication.

Re: Thank you very much for your comments regarding our manuscript. We revised our manuscript according to your comments. The followings are our reply to the points that you mentioned.

Major issues:

  1. Supplementary Figure 1 — there are faint bands between 300-400 bp in samples 8-3 and 8-5. Authors are encouraged to provide qPCR amplification curves for false-positive samples in separate supplementary figures. Observed Cq for qPCR-gG could be artifacts or linear increase of fluorescence.

Re: The amplicon length expected after specific amplification by nested PCR is 359 bp. The positions of the samples No.8-3 and 8-5 in Supplementary Figure 1 were around 300 bp and slightly lower than those of No.8-14 and 8-17 indicating the expected size. Therefore, we considered them as false-positive. According to your suggestion, we added Supplementary Figure 2 showing amplification curves of the samples No.8-3 and 8-5, and No.8-14 and 8-17, respectively.

  1. Authors mentioned that a considerable percent of macaques is BV-seropositive. However, using oral swabs only 2 out of 97 animals were positive after qPCR and LAMP testing. Thus, a question arises about seropositivity of the studied macaques and were there any infection symptoms false-positive and true-positive samples macaques? It can be assumed that much more animals were actually infected but were tested as negative because the infection did not clinically manifest at the time when samples were taken. In that sense, usage of NAATs for macaques can be limited and serological analysis would provide more reliable information.

Re: As we described in the introduction, B virus infections in macaques are mostly inapparent and induce latent infection. In addition, clinical symptoms are not always caused by viral reactivation in the monkey latently infected with BV. Thai group previously showed that 70-90% of cynomolgus monkeys in the regions where we captured the monkeys in this study were seropositive for BV (unpublished data), although we have not demonstrated a serological test for the samples in this study. As described in lines 70-73, serological test to detect anti-BV antibody shows the history infected with BV in the past, but not current shedding of the virus. Therefore, we cannot correctly evaluate the risk of the transmission of BV from the monkey by serological analysis. To determine whether the monkey is shedding BV, we have to detect BV or viral component. For this purpose, we developed qPCR and LAMP assays to detect viral genome in this study.

 Minor issues:

1.Minor language issues need to be corrected.

Re: Thank you for your suggestion. We corrected several points.

2.Page 3, line 127: “0.9 µM forward and reverse primers and 0.2 µM probe” — plausibly, these are not 10x but final concentrations.

Re: As you pointed out, the concentration of primers are the final concentrations. We corrected them.

3.Page 3, line 140: “that recognize six different sequences” — FIP and BIP primers are complementary each to 2 different sites.

Re: The primer set includes 6 primers targeting 8 different sites, as shown in Fig. 3 (F3, B3, FIP, BIP, LF, and LB). We corrected it to “that recognize eight different sites”.

4.Page 4, line 191: “The mixed nucleotide probe was used in the qPCR” — it could be seen on Figure 2 that a conservative region without any polymorphisms is located upstream of the designed probe. Plausibly, this region would be more beneficial for probe design.

Re: We attempted to design a probe in such conserved regions without any polymorphism. However, Tm value and GC % were not optimized as a qPCR probe.

Reviewer 2 Report

Amano et al. present their work on the development of qPCR and LAMP assays for detecting Herpes B virus in macaques. Herpes B virus is endemic in macaques, typically causing no clinical signs in them, but transmission to humans can result in fatal central nervous system infections. Both laboratory animals and wild macaques pose a significant risk to human health. Currently, Herpes B virus infection in primates is mainly confirmed through time-consuming virus isolation or the detection of antibodies using ELISA, and less frequently via PCR. Amano et al. have established qPCR and LAMP assays with high sensitivity for detecting different Herpes B virus strains in a short timeframe.

The paper is well-written and addresses an important topic, considering macaques are frequently used as animal models in infection studies. However, there are several points that need clarification to enhance the comprehensiveness and clarity of the paper.

11.       The paper mentions the collection of oral swabs and blood samples, with both qPCR and LAMP assays performed using oral swab samples. It is unclear what the blood samples have been used for in the study.

22.       The sampling procedure necessitated anesthesia for the macaques. Have any attempts been explored to facilitate sample collection and Herpes B virus analysis without direct interventions with the animals, such as obtaining urine or saliva from their environment or food sources?

33.       The authors report that 17 samples tested false-positive via qPCR targeting gG. It would be beneficial if the authors could provide an explanation for these false-positive results. Additionally, it would be interesting to know if similar observations have been documented in the literature when gG was targeted via PCR.

44.       Has the LAMP assay been tested for reactivity against other herpesviruses, especially HSV-1 and -2? This information is crucial when considering the application of LAMP assays to human samples in the future.

Author Response

Amano et al. present their work on the development of qPCR and LAMP assays for detecting Herpes B virus in macaques. Herpes B virus is endemic in macaques, typically causing no clinical signs in them, but transmission to humans can result in fatal central nervous system infections. Both laboratory animals and wild macaques pose a significant risk to human health. Currently, Herpes B virus infection in primates is mainly confirmed through time-consuming virus isolation or the detection of antibodies using ELISA, and less frequently via PCR. Amano et al. have established qPCR and LAMP assays with high sensitivity for detecting different Herpes B virus strains in a short timeframe.

The paper is well-written and addresses an important topic, considering macaques are frequently used as animal models in infection studies. However, there are several points that need clarification to enhance the comprehensiveness and clarity of the paper.

Answer: Thank you very much for your comments regarding our manuscript. We revised our manuscript according to your comments. The followings are our reply to the points that you mentioned.

1.The paper mentions the collection of oral swabs and blood samples, with both qPCR and LAMP assays performed using oral swab samples. It is unclear what the blood samples have been used for in the study.

Answer: Blood samples were used only for the health check, but not for the research. We clearly described it in the revised manuscript (Line 103).

2.The sampling procedure necessitated anesthesia for the macaques. Have any attempts been explored to facilitate sample collection and Herpes B virus analysis without direct interventions with the animals, such as obtaining urine or saliva from their environment or food sources?

Answer: Urine and saliva were not collected from the environment or foods, since it would be difficult to distinguish the individual. Additionally, once the urine was contaminated in the environment, it was more unlikely that the virus could be detected. In this study, No. 8-14 and 8-17, which were positive for BV in qPCR and LAMP using oral swab samples, were also tested using urine samples. But the results were negative in both qPCR and LAMP.

3.The authors report that 17 samples tested false-positive via qPCR targeting gG. It would be beneficial if the authors could provide an explanation for these false-positive results. Additionally, it would be interesting to know if similar observations have been documented in the literature when gG was targeted via PCR.

Answer: We rewrote the paragraph to explain it (Lines 215-226).

To date, there are some reports on BV detection using gG-qPCR primers [Ref A,B]. However, since the samples used in the previous reports were derived from M. mulatta, there should be no mutations in the primer-binding regions, and it is thought that false positive results did not occur. On the other hand, since our specimens were derived from M. fascicularis, it was possible that false positives were obtained.

Ref: A.    Perelygina, L.; Patrusheva, I.; Manes, N.; Wildes, M. J.; Krug, P.; & Hilliard, J. K. Quantitative Real-Time PCR for Detection of Monkey B Virus (Cercopithecine Herpesvirus 1) in Clinical Samples. J. Virol. Methods 2003, 109, 245–251. https://doi.org/10.1016/S0166-0934(03)00078-8.

  1. Wang, W.; Qi, W.; Liu, J.; Du, H.; Zhao, L.; Zheng, Y.; Wang, G.; Pan, Y.; Huang, B.; Feng, Z.; Zhang, D.; Yang, P.; Han, J.; Wang, Q.; & Tan, W. First Human Infection Case of Monkey B Virus Identified in China, 2021. China CDC Wkly. 2021, 3, 632–633. https://doi.org/10.46234/ccdcw2021.154.

4.Has the LAMP assay been tested for reactivity against other herpesviruses, especially HSV-1 and -2? This information is crucial when considering the application of LAMP assays to human samples in the future.

Answer: Although we have not examined our LAMP assay for human herpesviruses, we compared the primer sequences of the LAMP-UL27 with HSV-1 and HSV-2 reference sequences. We added Supplementary Fig. 3 to show the result. As shown in Supplementary Fig. 3, the sensitivities of the assay to HSV-1 and HSV-2 must be low due to the presence of mutations at the 3' terminus of FIP and BIP primers. We added sentences in lines 357-360.